# Epigenetic Induction of Secondary Metabolites Production in Endophytic Fungi *Penicillium chrysogenum* and GC-MS Analysis of Crude Metabolites with Anti-HIV-1 Activity

**DOI:** 10.3390/microorganisms11061404

**Published:** 2023-05-26

**Authors:** John P. Makhwitine, Hezekiel M. Kumalo, Sizwe I. Ndlovu, Nompumelelo P. Mkhwanazi

**Affiliations:** 1Discipline of Medical Microbiology, School of Laboratory Medicine and Medical Sciences, College of Health Science, University of KwaZulu-Natal, Durban 4000, South Africa; 220016491@stu.ukzn.ac.za; 2Drug Research and Innovation Unit, Discipline of Medical Biochemistry, School of Laboratory Medicine and Medical Science, University of KwaZulu-Natal, Durban 4000, South Africa; kumaloh@ukzn.ac.za; 3Department of Biotechnology and Food Technology, Faculty of Science, University of Johannesburg, Johannesburg 2028, South Africa; 4HIV Pathogenesis Programme, Doris Duke Medical Research Institute, School of Laboratory Medicine and Medical Sciences, University of KwaZulu-Natal, Durban 4000, South Africa

**Keywords:** human immunodeficiency virus, endophytic fungi, epigenetic modifiers, biosynthetic gene clusters, secondary metabolites

## Abstract

The continuous burden of human immunodeficiency virus-1 in Sub-Saharan Africa, coupled with the inability of antiretroviral agents to eradicate HIV-1 from viral reservoirs, the potential risks of drug resistance development, and the development of adverse effects, emphasizes the need to develop a new class of HIV-1 inhibitors. Here, we cultivated four endophytic fungal isolates from a medicinal plant, *Albizia adianthifolia* with the addition of small epigenetic modifiers, sodium butyrate, and valproic acid, to induce the expression of biosynthetic gene clusters encoding active secondary metabolites with probable anti-HIV activities. We identified a non-toxic crude extract of the endophytic fungus *Penicillium chrysogenum* treated with sodium butyrate to possess significantly greater anti-HIV activity than the untreated extracts. *Penicillium chrysogenum* P03MB2 showed anti-HIV activity with an IC_50_ of 0.6024 µg/mL compared to untreated fungal crude extract (IC_50_ 5.053 µg/mL) when treated with sodium butyrate. The profile of secondary metabolite compounds from the bioactive, partially purified extracts were identified by gas chromatography-mass spectrometry (GC-MS), and more bioactive compounds were detected in treated *P. chrysogenum* P03MB2 fractions than in untreated fractions. Pyrrolo[1,2-a]pyrazine-1,4-dione, hexahydro (13.64%), cyclotrisiloxane, hexamethyl (8.18%), cyclotetrasiloxane, octamethyl (7.23%), cyclopentasiloxane, decamethyl (6.36%), quinoline, 1,2-dihydro-2,24-trimethyl (5.45%), propanenitrile (4.55%), deca-6,9-diene (4.55%), dibutyl phthalate (4.55%), and silane[1,1-dimethyl-2-propenyl)oxy]dimethyl (2.73%) were the most abundant compounds. These results indicate that treatment of endophytic fungi with small epigenetic modifiers enhances the secretion of secondary metabolites with stronger anti-HIV-1 properties, acknowledging the feasibility of epigenetic modification as an innovative approach for the discovery of cryptic fungal metabolites which can be developed into therapeutic compounds.

## 1. Introduction

Highly active antiretroviral therapy (HAART) has significantly reduced viral load and improved life expectancy patient survival [1]. However, this therapy has not been able to sufficiently inhibit or terminate HIV-1 since it is administered after infection is established. HAART is often associated with various side effects and the emergence of drug-resistance in HIV strains [2,3]. On this account, it is essential to discover innovative and inexpensive anti-HIV agents from natural sources that may have fewer or no adverse effects with novel mechanisms [4].

Several bioactive compounds from endophytic fungi have shown the potential to inhibit HIV-1 enzymes and proteins involved in the HIV-1 life cycle [5]. Some fungal compounds explored as anti-HIV agents include altertoxins from *Alternaria tenuissima* and phenylspirodrimanes from *Stachybotrys chartarum* [6,7]. These compounds were found to inhibit a reverse transcriptase enzyme responsible for constructing a complementary deoxyribonucleic acid (DNA) from a ribonucleic acid (RNA) template [6,8]. The integrase enzyme, which is responsible for the insertion of HIV DNA into the host genome during the DNA recombination mechanism, is yet another HIV-1 target inhibited by fungal compounds [9]. Singh et al. [10] showed that xanthoviridicatins E and F derived from *Penicillium chrysogenum* are potent inhibitors of the HIV-1 integrase enzyme. Both compounds inhibited the cleavage process of HIV-1 integrase.

Apart from the HIV-1 targets, in a review by Roy et al. [5], fungal compounds were also identified to target the protease and glycoprotein 120 in the HIV-1 genome [5,11,12,13,14]. These studies indicate that fungi host structurally diverse molecules that can potentially target different sites in the HIV genome. However, there are few reports regarding the mechanisms of these compounds during HIV-1 replication [15]. One of the limitations is that fungal compounds rarely enter clinical trials, and this is due to their complex physiology and limited availability of genetic tools for manipulating their biosynthetic pathways to exploit their diverse structural repertoire [16].

Advances in genomics and innovative culturing methods have made it possible to revisit microbial-derived natural products after they were abandoned due to the increasing rediscovery of already known molecules [17]. The availability of high-throughput fungal genome sequences has revealed that many fungal biosynthetic gene clusters (BGCs) responsible for the biosynthesis of bioactive secondary metabolites are silent or transiently expressed when fungi are cultivated under laboratory conditions [18]. The proximal location of fungal BGCs to the telomers is believed to result in the silencing of these genes. In these regions, fungal BGCs are regulated by posttranslational modification markers, such as histone deacetylation, DNA methylation, or phosphorylation which results in the rearrangement of the chromatin, determining the accessibility of the DNA to transcription factors [19,20]. Therefore, synthetic media used to cultivate fungi in the laboratory often fail to stimulate the expression of these BGCs due to limited knowledge of the regulatory circuits and environmental stimulus responsible for their expression in the natural environment [18,21].

Recent introduction of innovative culturing methods, in this instance through incorporating small molecular epigenetic modifiers in fungal fermentations, have shown considerable successes in eliciting the expression of silent or transiently expressed fungal BGCs, yielding a plethora of bioactive compounds [22]. Epigenetic modifiers are small molecular compounds that target the epigenetic processes, such as inhibiting the transfer of methyl, acetyl, or alkyl groups, and thus lead to epigenetic alterations [22]. Two classes of small epigenetic modifiers, namely, histone deacetylases (HDACs) and DNA methyltransferases (DNMTs) inhibitors are typically used to spike fermentation broths during fungal cultivation. The HDAC inhibitors that have shown considerable success in inducing biosynthetic pathways in filamentous fungi include compounds, such as suberoylanilide (SAHA), sodium butyrate, and valproic acid. For example, treatment of *Aspergillus fumigatus* with valproic acid resulted in a 10-fold increase in fumiquinazoline C production [22]. Valproic acid act by inhibiting class 1 and 2 histone deacetylases (HDACs) and lead to the induction of proteosomal degradation of class 2 HDACs [20]. Exposure of *Streptomyces coelicolor* to sodium butyrate activated the biosynthetic pathway for the production of actinorhodin, which was otherwise silent in the uninduced strain [23,24]. Treatment of *Pestalotiopsis crassiuscula* with a DNMT inhibitor, 5-azacytidine, enhanced the production of the compound isosulochrin [25,26]. More examples of fungal secondary metabolites have been shown to be induced by treatment with epigenetic modifiers, indicating that these small compounds are efficient tools for elucidating biosynthetic pathways that are silent or transiently expressed in fungal species [27,28,29,30,31,32].

In this study, we have assessed the effects of small molecular epigenetic modifiers in activating silent BGCs in endophytic fungi *Penicillium chrysogenum* isolated from *Albizia adianthifolia* followed by a comparative analysis of anti-HIV activity of produced secondary metabolites between the treated and the untreated crude/fractionated extracts from the fungal culture. Furthermore, we performed GC-MS analysis on fractionated *P. chrysogenum* crude extracts. Sodium butyrate-treated extracts revealed increased anti-HIV activity and increased production of bioactive secondary metabolites compared to untreated extracts. These findings show a promise toward epigenetic modification as a powerful strategy to induce silent BGCs of endophytic fungi in order to produce secondary metabolites with anti-HIV properties.

## 2. Materials and Methods

### 2.1. Endophytic Fungi Isolation

Leaves and bark of *A. adianthifolia* (Flat crown) were harvested from the South Coast area of the eThekwini Municipality, KwaZulu-Natal. The plant samples were submitted for taxonomic identification and registration (voucher number 18232) at the University of KwaZulu-Natal (UKZN), School of Life Science Herbarium. A method outlined by Arivudainambi et al. [33] and Petrini [34] was followed to isolate five endophytic fungi from the leaves and bark of *A. adianthifolia*. Briefly, the plant samples (leaves and barks) were properly washed with distilled water to eliminate epiphytes. The surfaces were then sterilized in 70% (*v*/*v*) ethanol (Lichro chemicals, Durban, South Africa) for 1 min, and thereafter dipped in sodium hypochlorite 1% (*v*/*v*) (Novachem, Quito, Ecuador) for 3 min before dipping them once more in 70% (*v*/*v*) ethanol for 1 min. The last wash was performed three times in distilled water and air-dried. The leaves and barks were cut into small pieces (5–7 mm), and aseptically added onto potato dextrose agar (PDA, Neogen, Lansing, MI, USA) and malt extract agar (MEA, Neogen, Lansing, MI, USA) plates supplemented with 100 μg/mL ampicillin (Thermo Fischer, New York, NY, USA) to inhibit bacterial growth. The cultures were incubated in the dark at 25 °C for 5–7 days, followed by subsequent sub-culturing until single fungal cultures were retained and preserved in 25% glycerol at −80 °C for future purposes.

### 2.2. Treatment of Endophytic Fungi with Small Epigenetic Modifiers

Small-molecule modifiers, sodium butyrate, and valproic acid (Sigma-Aldrich, Johannesburg, South Africa) were diluted in distilled water and added to 10 mL of malt extract (ME) broth at a final concentration of 25 µM inoculated with three small pieces (1 cm^2^) of fungal mycelia [35]. The fungal cultures (treated and untreated) with small-molecule modifier were incubated in the dark at 25 °C for 14 days. Ten milliliters of fungi-free PD broth (control) were treated with small-molecule modifiers (valproic acid and sodium butyrate) added to final concentrations of 25 µM. The culture controls were treated in the same way as the experimental controls to determine whether the small-molecule modifiers and solvent altered the observed activity in the absence of fungal inoculum. The assays were performed in duplicates.

### 2.3. Crude Extraction of Fungal Secondary Metabolites

The fungal secondary metabolite crude extracts were prepared as we have previously reported in Nzimande et al. [36] with minor modifications. Briefly, after 14 days of incubation, an equal volume of absolute methanol (ChemLab supplies, Johannesburg, South Africa) was added to the whole fungal culture and incubated overnight with shaking on a rotary shaker (Reflecta Laboratory supplies, Germiston, South Africa) at 150 rpm. The mycelia were then separated using gauze and the culture supernatant was transferred into a centrifuge tube. The retained supernatant was evaporated at 40 °C to dry the extracts. The dried crude extracts were stored at −80 °C until further use. Before each use, the extracts were resuspended in distilled water to a concentration of 300 μg/mL. The following day, gauze was used to separate the mycelia from the supernatant (crude extracts). Before each use, the dried extract was diluted in distilled water to a final concentration of 300 µg/mL prior to antiviral activity screening.

### 2.4. Molecular Identification of Endophytic Fungi of Active Fungal Extracts

#### 2.4.1. Cultivation and DNA Isolation of Endophytic fungi

The fungal isolate (strain P03MB2) was cultured in the dark on solid malt extract agar (MEA) for 5–7 days at 25 °C. For genomic DNA extraction, 100 mg of mycelia from the fresh fungal hyphae was collected. The Norgen Plant/Fungi DNA isolation kit (25240, Norgen Biotek, Thorold, ON, Canada) was used to extract genomic DNA following the manufacturer’s instructions. The extracted DNA was evaluated for its purity and concentration using 1% agarose and a NanoDrop 2000c Spectrophotometer (Thermo Scientific, Wilmington, NC, USA), respectively.

#### 2.4.2. Amplification of Fungal DNA

The internal transcribed spacer (ITS) sequence region of the extracted fungal DNA was amplified using primers, forward primer, ITS1 (5′-TCCGTAGGTGAACCTGCGG-3′) and reverse primer, ITS4 (5′-TCCTCCGCTTATTGATATGC-3′). The polymerase chain reaction (PCR) master mix was made up of Phusion hot start II high fidelity polymerase (Thermo-Scientific, Carlsbad, CA, USA) (1×), ITS1 primer (0.5 μM), ITS4 primer (0.5 μM), 50 ng DNA, and nuclease-free water (9 μL). The PCR was carried out under the subsequent conditions: Initial denaturation at 98 °C for 30 s, denaturation at 30 cycles of 98 °C for 10 s, annealing at 45 °C for 15 s, extension at 72 °C for 15 s, and final extension at 72 °C for 2 min. The PCR products were evaluated on 1% agarose gel and purified using the PureLink^®^ PCR Purification kit, Invitrogen (Thermo Scientific, Carlsbad, CA, USA) following the manufacturer’s instructions. The purified amplicons for each fungal isolate were measured using NanoDrop 2000c.

#### 2.4.3. Identification of Endophytic Fungi by ITS Sequencing

The amplicons were sent to Central Analytical Facility (CAF) at Stellenbosch University (Stellenbosch, South Africa) for ITS region sequencing. The quality assessment of the ITS sequences was performed using Snap gene Viewer (version 6.0.6). Subsequently, the quality of ITS sequences was compared using Basic Local Alignment Tool (BLAST) application of the nucleotide database of the National Centre for Biotechnology Information (NCBI) with gene databases. The isolate P03MB2 was identified as *Penicillium chrysogenum* and the consensus sequence was deposited on NCBI (accession number ON989875). The sequences with the highest hits were chosen to develop the phylogenetic tree using the maximum-likelihood algorithm in MEGA X [37,38]. Evolutionary distances were computed using the Tamura-Nei model [38] and these were used to determine evolutionary distances. The percentage of replicate trees and tree robustness was evaluated using a bootstrap test (1000 replicates) [39].

### 2.5. Cell Cultures

The TZM-bl and HEK 293T cell lines were obtained from the NIH AIDS research and reference reagents program. The TZM-bl and HEK 293T are Hela cell lines that express large amounts of CD4 and CCR5. They were initially generated from JC.53 cells by introducing separate integrated copies of the luciferase gene under the control of the HIV-1 promoter [37,38,39]. The TZM-bl cell lines were chosen based on their high susceptibility to infection with most strains of HIV. TZM-bl cell lines were cultured in high-glucose Dulbecco’s Modified Eagle’s medium (DMEM) (Lonza Inc., Walkersville, MD, USA) containing 25 mM HEPES buffer, 10% FBS, and 50 µg/mL gentamycin. Both cell lines were incubated at 37 °C with 5% CO_2_.

#### 2.5.1. Transformation of XL1 Blue Super Competent Cells and Construction of Viral Plasmids

For transformation, 1 µL of plasmid DNA (pNL4.3 Luc, VSV-g, and CMV) was added to 50 µL of super competent cells to a pre-chilled Luria-Bertani (LB) broth (Neogen, MI, USA) The transformation reaction was gently mixed and incubated on ice for 30 min. Thereafter, the reactions were placed on heat pulse for 45 s at 42 °C, followed by another incubation on ice for 2 min. Pre-heated super optimal broth with catabolite repression (S.O.C) medium (Thermo Fischer, Randburg, South Africa) at 42 °C was added to the transformation solution and incubated at 37 °C for 1 h, with shaking on a rotary evaporator at 230 rpm. Transformation reactions were plated on nutrient agar plates supplemented with 100 µg/mL ampicillin, then incubated at 37 °C for 16 h. After incubation, the colonies were transferred to the master plate using pipette tips. Then, the tips were dislodged into 15 mL of pre-heated LB broth supplemented with 100 µg/mL ampicillin and placed in a shaking incubator at 37 °C at 230 rpm.

#### 2.5.2. Plasmid DNA Purification

The GeneJET plasmid miniprep kit (Thermo Scientific, Vilnius, Lithuania, EU) was used to purify plasmid DNAs following the manufacturer’s instructions. The DNA quality was assessed using NanoDrop™ 2000 software Version 1.6.198 (Thermo Scientific, Randburg, South Africa), followed by 1% agarose gel electrophoresis.

#### 2.5.3. Generation of Pseudoviruses by Transfection

HIV-1 NL4.3 stocks were generated by transfection of HEK 293T cells. Briefly, 1 day before transfection 3 × 10^6^, 293T cells/mL were seeded in a T-75 culture flask with 5 mL of supplemented DMEM to achieve 50–80% confluency of monolayers. The transfection solution containing 12 µL of Env plasmid DNA pCMV-VSV-g and 12 µL of backbone plasmid DNA pNL4.3 GFP-VSV-g in DMEM was transfected using 45 µL of XtremeGene (Promega, Southampton, UK) following the manufacturer’s instructions. After 48 h of incubation, the transfection medium was replaced with a fresh medium. The supernatant was harvested and filtered after 48 h, and aliquots were stored at −80 °C until use.

#### 2.5.4. Titration of Virus (TCID Assay)

The TCID_50_ of the generated HIV-1 NL4.3 stocks was determined. Briefly, TZM-bl cells prepared at a density of 1 × 10^5^ cells/well in DMEM (supplemented with 25 mM HEPES buffer, 10% FBS, and 50 µg/mL gentamycin) containing 37.5 mg/mL DEAE-dextran were cultured in a 96-well plate. TZM-bl cells were infected with a serial dilution of the virus stock in duplicate, starting with a 1/10 dilution. After 48 h of incubation, the culture medium (100 µL) was replaced with fresh DMEM media, and 100 µL Bright-Glo reagent (Promega, Madison, WI, USA) was added to each well under low light conditions. Complete cell lysis was achieved following incubation of the plate at room temperature for 2 min. All of the contents were transferred to a corresponding 96-well bottom flat black plate (Corning Incorporated, Glendale, AZ, USA). The relative luminescence was measured immediately using a Victor 2 luminometer plate reader (Perkin-Elmer Life Sciences, Shelton, CT) at 540 nm. The TCID_50_ was determined by the virus dilution that was able to elicit 50,000 relative light units (RLUs). A standard amount of virus (50,000 RLUs; multiple of infection (MOI) = 0.05) was used in the replication and drug susceptibility assays and the level of viral replication was expressed as a percentage of the RLUs.

### 2.6. Evaluation of Cytotoxicity of Fungal Crude Extracts

MTT (3-[4,5-dimethylthiazol-2-yl]-2,5-diphenyltetrazolium bromide) cell proliferation assay kit (Thermo Scientific, Randburg, South Africa) was used to evaluate the cytotoxicity of fungal crude extracts and compounds used in this study, following the manufacturer’s instructions. Briefly, 10 µL of 300 µg/mL treated and untreated fungal crude extracts with epigenetic modifiers and standard drug Zidovudine (AZT) at 300 µg/mL (NHI AIDS research and reagent program) were serially diluted 10-fold from 300 μg/mL in DMEM containing 10% heat-inactivated fetal bovine serum, 50 μg/mL gentamycin, and 25 mM HEPES buffer in a 96-well plate. Azidothymidine (AZT) at 300 μg/mL was used as the positive control and uninfected cells as the negative control. Wells without any drug/crude extract were used as a negative control. TZM-bl cell suspension prepared at a density of 1 × 10^5^ cells/mL in DMEM containing 37.5 µg/mL diethyl aminoethyl (DEAE)-dextran were seeded in the 96-well plate and incubated at 37 °C, 5% CO_2_ for 48 h. After incubation, 12 mM of MTT stock solution (10 µL) was added to each well and incubated for 4 h. Thereafter, the media was replaced with a fresh DMEM medium. Formazan crystals were solubilized by adding 50 µL of diluted DMSO (0.2%) and incubated for 10 min. The absorbance was read at 540 nm using an ELISA plate reader (PerkinElmer, Waltham, MA, USA). The percentage cell viability was calculated using the following formula:(%)Cell Viability = ((Sample absorbance − Cell − free sample blank))/((Mean media control absorbance)) × 100

The 50% cytotoxic concentration (CC_50_) causing visible morphological changes in 50% of TZM-bl cells, with respect to cell control, were determined using GraphPad Prism Software (version 5) [40].

### 2.7. Luciferase-Based Antiviral Assay

A luciferase-based antiviral assay was performed based on the method by Nutan et al. [41] using TZM-bl cell lines [42,43,44]. In this experiment, the crude extracts (sodium butyrate treated and untreated) and AZT control were serially diluted to achieve a concentration range of 3.0 × 10^−7^ to 300 µg/mL in DMEM containing 10% heat-inactivated fetal bovine serum, 50 µg/mL gentamycin, and 25 mM HEPES in a 96-well flat-bottom plate. Fifty microliters of diluted HIV-1 NL4.3 (MOI = 0.05) virus was dispensed to all the wells except for cell growth control (uninfected and untreated) wells and incubated for 1 h at 37 °C, 5% CO_2_. TZM-bl cells, prepared at a density of 1 × 10^5^ cells/mL in DMEM containing 37.5 mg/mL DEAE-dextran, were seeded in all wells in a 96-well plate and incubated at 37 °C, 5% CO_2_ for 72 h. After incubation, the culture medium (100 µL) was replaced with a new DMEM media, and 100 µL Bright Glo reagent (Promega, Madison, WI, USA) was added to each well under low light conditions. Complete cell lysis was achieved following incubation of the plate at room temperature to a corresponding 96-well bottom flat black plate (Corning Incorporated, Glendale, AZ, USA). The relative luminescence unit was measured immediately using a Victor 2 luminometer plate reader (Perkin-Elmer Life Sciences, Shelton, CT, USA) at 540 nm. The percentage of viral inhibition was expressed using the following equation:% HIV inhibition = ((Average sample − Average control))/([1 − (Average viral control − Average control)]) × 100

The half-maximal inhibitory concentration (IC_50_) causing morphological changes in the HIV inhibition dose-response curve by 50% was calculated using GraphPad Prism Software (version 5).

### 2.8. Bioassay-Guided Fractionation and Chemical Profiling of Secondary Metabolites Using Gas Chromatography-Mass Spectrometry (GC-MS)

#### 2.8.1. Extraction of Secondary Metabolites through Large-Scale Fermentation

The secondary metabolite extraction was carried out as we have outlined above. Briefly, the endophytic fungi that exhibited bioactivity in preliminary screening following the luciferase-based assays were cultured (six fungal plugs of 1 mm^2^ each) in 500 mL Erlenmeyer flasks containing 150 mL of malt extract broth at 25 °C for 14 days, with shaking on a rotary evaporator at 150 rpm. After incubation, an equal volume of absolute methanol was added to the whole fungal culture and incubated overnight with shaking on a rotary shaker at 150 rpm. The mycelia was then separated using gauze and the culture supernatant was transferred into a centrifuge tube. The retained supernatant was evaporated at 40 °C to dry the extracts. The dried crude extracts were stored at −80 °C until further use.

#### 2.8.2. Solid-Phase Extraction (SPE) of Fungal Crude Extracts

Active crude extracts were fractionated as described by Stoszko et al. [45]. The dried crude extracts were sequentially fractionated using various cartridges (Waters Corporation, Prague, Czech Republic) following the manufacturer’s instructions. Briefly, lipids were removed by hexane/water extraction 50/50 (*v*/*v*), 4-mL total volume, and an aqueous phase was collected and dried. The dried material was reconstituted in 50% methanol (MeOH) to a concentration of 100 μg/mL. In addition, 1 mL of this solution was spiked with 20 µL of phosphoric acid (Sigma-Aldrich, South Africa) and loaded onto HLB (hydrophilic-lipophilic-balanced reverse phase), MCX (mixed-mode, strong cation-exchange), and MAX (mixed mode, strong anion-exchange) cartridges obtained from Waters Corporation (Prague, Czech Republic). The adsorbed compounds were desalted and stepwise eluted with increasing (5, 45, and 95%) organic solvent (MeOH), providing sample F5, F45, and F95 fractions, respectively. For the fractions obtained using the HLB cartridge (F5-HLB, F45-HLB, and F95-HLB), for fractions from the MCX cartridge (F5_MCX, F45_MCX, and F95_MCX), and for fractions from the MAX cartridge (F5_MAX, F45_MAX, and F95_MAX). Eluted fractions were concentrated by evaporation to dryness at 40 °C under a gentle stream of nitrogen. Dried fractions were reconstituted with 1 mL of acetonitrile and filtered with a 0.2 µm filter before use. HLB was based on N-vinylpyrrolidone–divinylbenzene copolymer. MCX was a cation-exchange sorbent that represents the HLB material modified with SO3H− groups. MAX was an anion-exchange cartridge. After each round of fractionation, all samples were tested for anti-HIV-1 activity using the luciferase-based antiviral assay described earlier.

#### 2.8.3. Gas Chromatography-Mass Spectrometry Analysis

A gas chromatography-mass spectrometry (GC-MS) analysis was performed to identify secondary metabolites from the partially purified crude fungal extracts. GC-MS was performed in the chemistry department at the University of KwaZulu-Natal (PMB). Briefly, GC-MS analysis was performed by splitless injection (spilt 20:80-8-200-5M-8-260-10M10-280-HP5-ETOH) of 1.0 µL of the sample in methanol on a Hewlett Packard 6890 (USA) gas chromatograph. The Agilent 19091S-433 column (30 m × 250 μm × 0.25 µm) was used to separate the samples. The starting column temperature was 35 °C with a hold time of 3 min. The temperature was set to increase at 8 °C/min, with a maximum temperature of 280 °C. One microliter of the sample was injected into the port, subsequently vaporized, and transported down the column utilizing helium as the carrier gas at a flow rate of 1 mL/min. At 70 eV, the MS spectrum was captured. Following the separation in the column, the components were identified and evaluated using a flame ionization detector (FID). Compounds were identified by comparing the spectrum of unknown compounds to the spectrum of known compounds in the National Institute of Standards and Technology (NIST MS 2.0) structural library to determine their names, molecular weight, and structure [46].

## 3. Results

### 3.1. Isolation and Molecular Identification of Endophytic Fungi

A total of four endophytic fungi (strains P03PB2, P03PB3, P03PL3, and P03MB2) were isolated from *A. adiantifolia*. The four fungal isolates were further treated with two epigenetic modifiers (12 μM valproic acid or 12 μM sodium butyrate) and only P03PL3 and P03MB2 showed considerable anti-HIV activity (Appendix A). The fungal isolate P03MB2 treated with sodium butyrate showed a 5-fold increase in anti-HIV-1 activity when compared with the untreated fungal culture and it was selected for further characterization. The basic local alignment tool (BLAST) analysis of the internal transcribed spacer (ITS) sequences revealed that strains P03MB2 shared 100% identity with *Penicillium chrysogenum* (Figure 1A). A phylogenetic tree of the endophytic fungi *P. chrysogenum* P03MB2 was constructed and revealed 100% relatedness to other *Penicillium* species (Figure 1B).

### 3.2. Cytotoxicity Effects of Crude Methanol Extracts

The crude extracts resuspended to 300 µg/mL in distilled water were first assessed for cytotoxicity against TZM-bl cell lines. The percentage of cell viability of TZM-bl cells against methanol crude fungal extracts treated and untreated with sodium butyrate is shown in Figure 2. The results revealed that all endophytic fungal extracts showed no cytotoxicity on TZM-bl cells with >80% cell viability and CC_50_ > 1000 µg/mL. The positive reference, AZT (300 µg/mL) did not show toxicity on TZM-bl cells with average cell viability of 106% and CC_50_ of 21.4 µg/mL. Figure 2 indicates the cytotoxic effects of the crude extracts with their cell cytotoxicity at 50%.

### 3.3. Luciferase-Based Antiviral Activity Assay on Crude Extracts

Antiviral effects of the crude extracts were measured by calculating the percentage of virus inhibition. The *P. chrysogenum* P03MB2 crude extract showed anti-HIV activity with a half-maximal inhibitory concentration (IC_50_) of 0.048 µg/mL with a fold-change of 5 compared to the untreated fungal extract (IC_50_ 0.009 µg/mL) (Figure 3). The untreated *P. chrysogenum* P03MB2 crude extract showed HIV-1 inhibition of up to 54%, while the treated crude extracts reached 100% inhibition. The positive reference, AZT, indicated a positive dose-dependent curve with IC_50_ of 0.033 µg/mL (Figure 3).

### 3.4. Bio-Assay Guided Fractionation Approach to the Anti-HIV Activity of Secondary Metabolite Fractions (Solid-Phase Extraction)

After a single round of fractionation of crude extracts, we generated six fractions per cartridge (three fractions from the treated and three fractions from the untreated) *P. chrysogenum* P03MB2, and three fungi-free sodium butyrate-treated fractions (control). All these generated fractions were screened for anti-HIV activity in a dose-response approach with initial concentration at 100 µg/mL. All three fungi-free sodium butyrate-treated fractions did not reveal any anti-HIV activity. *P. chrysogenum* P03MB2 fractions eluted with 45% MeOH from all three fractions (HLB 45%, MCX 45%, and MAX 45%) revealed anti-HIV activity (Figure 4), while all fractions eluted with 5% and 95% did not indicate potent anti-HIV activity. The observed anti-HIV activities showed 0.63- and 0.33-fold increase for MCX 45% and MAX 45% fractions, respectively when compared to the untreated extracts. The best anti-HIV activity was noted from the HLB 45% fraction with an IC_50_ of 0.6024 µg/mL compared to 5.053 µg/mL of the untreated extract, which was a remarkable fold-change of 8.39.

### 3.5. GC-MS Analysis

GC-MS analysis of methanolic (45%) extracts from *P. chrysogenum* isolate revealed the presence of different compounds as mapped from the NIST library [47]. The gas chromatogram of fractionated crude extract of treated and untreated *P. chrysogenum* indicated a variety of metabolites with different retention times per cartridge; the results are shown in Appendix A. A total of 110 compounds were detected from *P. chrysogenum*-treated and untreated fractions produced from HLB, MCX, and MAX cartridges (Table 1). The most interesting compounds include cyclobutane carbonitrile. A compound with this scaffold (1-(4-chlorophenyl) cyclobutane carbonitrile) is available commercially and has been reported as a potential treatment for HIV [48]. Pyrrolo[1,2-a]pyrazine-1,4-dione, hexahydro (13.64%), cyclotrisiloxane, hexamethyl (8.18%), cyclotetrasiloxane, octamethyl (7.23%), cyclopentasiloxane, decamethyl (6.36%), quinoline, 1,2-dihydro-2,24-trimethyl (5.45%), propanenitrile (4.55%), deca-6,9-diene (4.55%), dibutyl phthalate (4.55%), and silane, [(1,1-dimethyl-2-propenyl)oxy]dimethyl (2.73%). Compounds detected from sodium butyrate fractions were excluded from *P. chrysogenum* fractions as they were used as a negative control (Appendix A).

## 4. Discussion

In this study, we used epigenetic modifiers, sodium butyrate, and valproic acid to induce the expression of biosynthetic gene clusters. Endophytic fungi are a rich source of secondary metabolites with various bioactivities including antiviral. The secondary metabolites from the endophytic fungi include steroids, alkolaids, phenols, isocoumarins, xanthones, quinones, and terpenoids [82,83]. The fungal biosynthetic pathways encoding these metabolites are typically clustered near the telomeres and they are subject to tight regulation mainly driven by various epigenetic markers [20]. Therefore, these BGCs are usually silent when the fungal species is cultivated under standard laboratory conditions [84]. Previous studies have shown that small compounds, such as valproic acid and sodium butyrate can inhibit some of these epigenetic markers, leading to the expression of a plethora of secondary metabolites [85]. In this study, we isolated four endophytic fungi from the medicinal plant, *A. adianthifolia.* First, we screened the crude extracts from these fungal isolates for cytotoxicity against TZM-bl cell lines. Then, we observed that the TZM-bl cells maintained cell viabilities of ≥80% with CC50 > 1000 µg/mL after treatment with the fungal crude extracts, indicating that these extracts were not toxic to these cells. It was critical to evaluate the extracts’ cytotoxicity to eliminate the possibility of a direct inhibitory effect of the crude extracts to the TZM-bl cells, which might lead to an inaccurate conclusion [61]. The non-toxicity of these crude extracts to the TZM-bl cells used in this study also suggests their safety if they are considered for further evaluation as candidate drugs for HIV treatment. In this study, the MTT viability assay was used to measure the biochemical marker, to evaluate the metabolic activity of the cell, and to measure the concentration of the compounds/crude extracts that are cytotoxic to the cells.

During our preliminary activity screening, we determined that two fungal isolates (P03PB2 and P03MB2) showed anti-HIV-1 activity. After perturbation with the selected epigenetic modifiers, P03MB2 was selected for further characterization since it showed an increase in anti-HIV-1 activity of more than 8-fold upon treatment with sodium butyrate. A similar effect has been reported where valproic acid was added to the growth media of *Aspergillus fumigatus* [22]. The addition of valproic acid overexpressed genes encoding fumiquinazoline C and improved its production by 10-fold. The crude extract of P03MB2 without treatment with epigenetic modifiers had weak anti-HIV activity compared to the treated crude extracts. Sodium butyrate is an epigenetic modifier which inhibits the histone deacetylase activity, leading to the differentiation of the eukaryotic cells [86,87]. Previous studies showed that sodium butyrate treatment had the highest induction rate of 22 cryptic secondary metabolites from endophytic *Nigrospora sphaerica* [88]. The endophytic fungus isolate P03MB2 was then identified to share 100% homology with *Penicillium chrysogenum* based on the internal transcribed spacer (ITS) rRNA gene sequence analysis. The species with the genus *Penicillium* including *P. chrysogenum* are well-investigated for their ability to produce a wide range of bioactivities, including antibacterial and antifungal properties, but there have been few reports on their anti-HIV activities [89,90,91]. To cite a few compounds that have been described as anti-HIV compounds, Singh et al. [10] discovered xanthoviricandins E and F, indicating that *P. chrysogenum* might be a potential source of anti-HIV inhibitors. This study showed that more bioactivity potential could be revealed by using various BGC activation strategies, including treatment with epigenetic modifiers, such as sodium butyrate.

The crude extracts of the *P. chrysogenum* P03MB2 were then fractionated to simplify the complex extract using three distinct cartridges (HLB, MCX, and MAX) targeting neutral, acidic, and basic compounds, respectively. The adoption of the three cartridges was inspired by a study published by Stoszcko et al. [45], where they reported on the discovery of gliotoxin with HIV-1 latency reversal properties. The Oasis cartridges are the next generation of C18 reverse-phase prepacked columns with improved chemistry. They are designed for the extraction of a wide range of compound chemistries from various matrices. In addition, they provide excellent recoveries without the need for worrying about the drying of the sorbent matrix. We chose these cartridges based on their suitability for an untargeted discovery approach where we do not know the chemical properties of the bioactive compound. Only the fractions that retained anti-HIV activity were selected for chromatographic analysis. The GC-MS results of the active fractions demonstrated five compounds that have been previously shown to exhibit anti-HIV properties. These compounds include coumarin and 1-propyl-9-tetradecenoate from the treated fraction and trans-1-(-*p*-ethoxyriphenyl)-1-dodecen-3-one, trans-2-hexadecenoic acid, and methyl-10-trans-12-cis-octadecadienoate from the untreated fraction of MAX (Table 1). Our findings suggest that the fungal isolate reported here has a potential to produce anti-HIV compounds that could be considered as drug candidates for further development. These findings are in good agreement with previous studies describing *Penicillium sp.* as a producer of antiviral compounds against HIV-1.

Furthermore, the findings presented here suggest that hydrophilic compounds (targeted in the HLB cartridge) are responsible for the bioactivity observed from the crude fractions. Comparative profiling with GC-MS spectra showed an increased production of secondary metabolites from the *P. chrysogenum*-treated fraction than the untreated fraction produced from the HLB cartridge. The limitation of using GC-MS is that it only identifies the volatile compound and it is possible that other compound groups may be responsible for the observed bioactivities. In the future, high-resolution liquid chromatography-tandem mass spectrometry (HRLC-MS/MS) should be considered to comprehensively cover the chemical space of compounds that might be responsible for the observed bioactivities. Our study results suggest the possible activation of biosynthetic genes in *P. chrysogenum* P03MB2 by the addition of sodium butyrate due to the observed increase in anti-HIV activity and increased compound profile in the bioactive fraction spectra. However, more rounds of purification of the bioactive extracts and deep genome analysis of fungal species are recommended to identify the compound responsible for bioactivity and to gain insight into the differential expression of the encoding biosynthetic pathway upon perturbation by sodium butyrate.

## 5. Conclusions

In this study, the effects of epigenetic modifiers in activating silent BGCs of endophytic fungi *P. chrysogenum* P03MB2 for increased production of secondary metabolites as anti-HIV-1 inhibitors were investigated. Our results showed that *A. adianthifolia* harbors fungal endophytes with interesting bioactivities that remain unexplored. The fractionated *P. chrysogenum* P03MB2 extract revealed an increased anti-HIV activity. Fractions of treated fungal isolates produced from the HLB cartridge indicated an 8-fold increase in anti-HIV activity compared to the untreated fungal fractions. Furthermore, GC-MS analysis profile suggested an increased compound profile upon treatment with sodium butyrate, and thus indicate that this approach can give access to compounds that are otherwise silenced under standard laboratory cultivation conditions. The presented screening approach further suggests epigenetic modification as a powerful tool in screening for novel chemistries toward the discovery of new bioactive compounds.

## Figures and Tables

**Figure 1 microorganisms-11-01404-f001:**
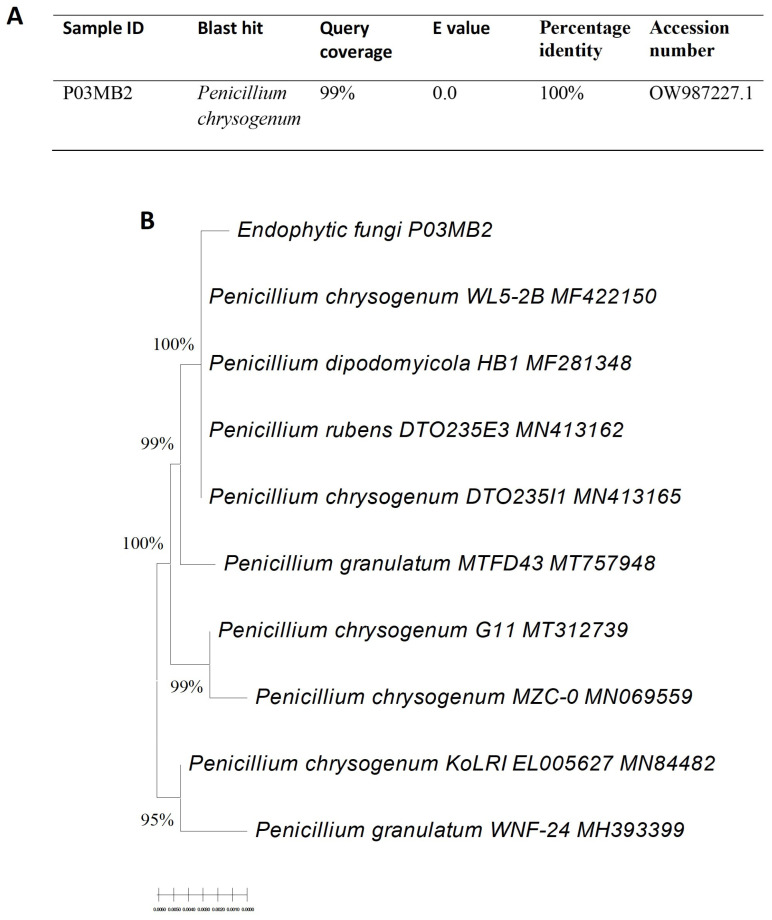
The maximum-likelihood tree was constructed based on ITS gene sequences of P03MB2, with closely-related fungal ITS sequences accessed from GenBank using the BLASTN tool. Panel (**A**) shows the similarity search of strain P03MB2, while Panel (**B**) shows the maximum likelihood of endophytic fungi *P. chrysogenum* P03MB2. The sequences were aligned using ClustalW. Bootstrap values included 1000 replicates using MEGA software (version 11.06) and are displayed on the tree branches.

**Figure 2 microorganisms-11-01404-f002:**
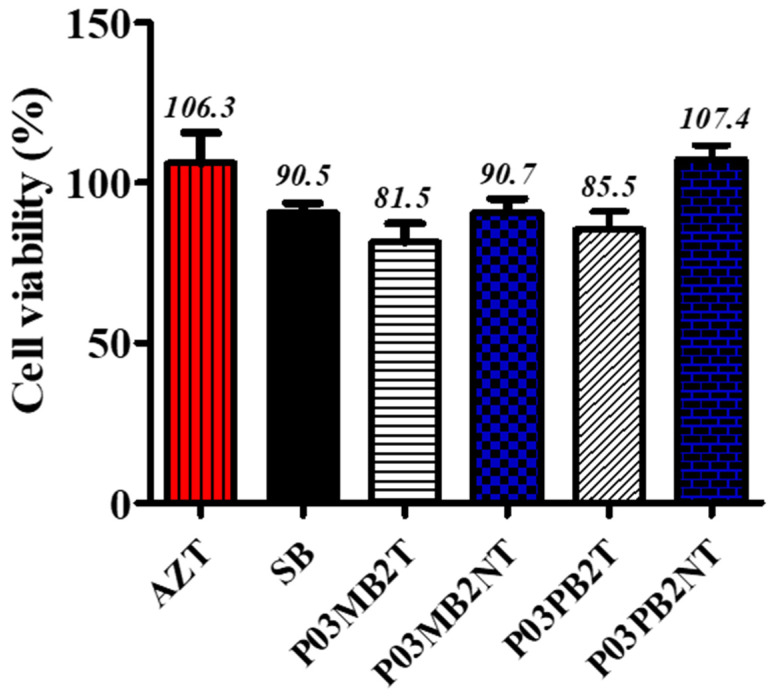
Percentage of cell viability of TZM-bl cell lines against methanol crude fungal extracts treated and untreated with sodium butyrate using the MTT viability assay. Results were obtained from two independent experiments; data represent the mean SEM. The *x*-axis shows the different treatments (treated or not treated crude extracts and a positive control, AZT (zidovudine), SB (sodium butyrate), PO3 (plant 3A, *A. adianthifolia*), M (malt extract agar), P (potato dextrose agar), B (bark), 2 (isolate number), T (treated extract), NT (not treated crude extract).

**Figure 3 microorganisms-11-01404-f003:**
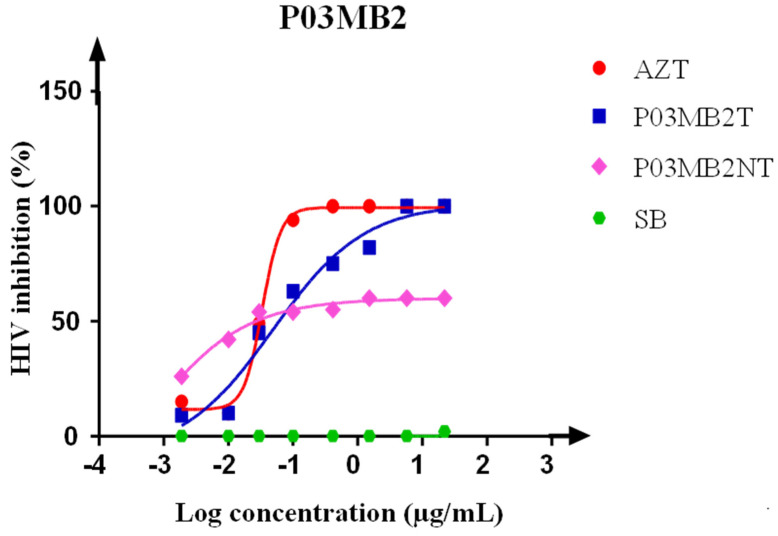
Dose-response curve showing anti-HIV-1 activity of sodium butyrate-treated crude extracts of *P. chrysogenum* P03MB2T (blue) and untreated crude extracts of *P. chrysogenum* P03MB2NT (pink) tested in TZM-bl cell lines. Fungi-free sodium butyrate (SB) crude extract (green) was included as a negative control, while AZT (red) was used as a positive control. The graph shows inhibitory concentration at 50% inhibition (µg/mL). The IC_50_ of P03MB2T was 0.048 µg/mL, P03MB2NT was 0.0090 µg/mL, and AZT (positive) control IC_50_ was 0.0330 µg/mL. The *x*-axis shows the serial dilution of crude extract concentrations in Log scale and the *y*-axis shows the percentage of HIV-1 inhibition.

**Figure 4 microorganisms-11-01404-f004:**
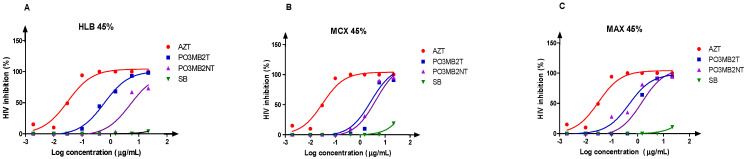
Dose-response curves of three crude extract fractions eluted in pre-concentration cartridges ((**A**) HLB 45%, (**B**) MCX 45%, and (**C**) MAX 45%). The sodium butyrate-treated *P. chrysogenum* P03MB2T (blue), untreated *P. chrysogenum* P03MB2NT (pink), AZT positive control (red), and fungi-free sodium butyrate-treated crude extract fraction (SB) as negative control (green). The *x*-axis shows serial dilution of crude extract concentrations in the Log scale and the *y*-axis shows the percentage of HIV-1 inhibition. The anti-HIV activity of fractionated crude extracts was tested against the HIV-1 infected TZM-bl cells. In Panel A (HLB 45%: *P. chrysogenum* PO3MB2T, IC_50_ 0.6024 µg/mL, *P. chrysogenum* PO3MB2NT, IC_50_ 5.053 µg/mL); Panel B (MCX 45%: *P. chrysogenum* PO3MB2T, IC_50_ 4.246 µg/mL; *P. chrysogenum* PO3MB2NT, IC_50_ 2.707 µg/mL); Panel C (MAX 45%: *P. chrysogenum* PO3MB2T, IC_50_ 1.456 µg/mL, *P. chrysogenum* PO3MB2NT, IC_50_ 0.4809 µg/mL).

**Table 1 microorganisms-11-01404-t001:** GC-MS compounds identified from *P. chrysogenum* P0MB2-treated and untreated fractions produced from HLB 45%, MCX 45%, and MAX 45% cartridge and their retention time, peak area, and biological properties.

Similarity Index	Retention Time (min)	Compound Name	Associated Biological Activity	Natural Product Sources	Cartridge	References
95	3.643	Propanenitrile	antimicrobial	*Brassica rapa*	MCX; MAX and HLB	[49,50]
91	3.6625	Propargylamine	Transcription repressing	*Alternaria alternata*	MCX; MAX and HLB	[36,51]
89	3.540	Cyclobutane carbonitrile	Anti-inflammatory; anti-depressant; anti-cancer and antiviral	*Agelas sceptrum* (sceptines) of the sea sponge	MCX; MAX and HLB	[52,53]
89	6.625	2-Butyne	antimicrobial	NR	MCX; MAX and HLB	[54]
64	4.070	Cyclotrisiloxane, hexamethyl	Antioxidant properties; antibacterial activity	*P. chrysogenum*; Extract of *Acacia karoo*; *Olea europaea* L. West Anatolian olive leaves. Red seaweed *Kappapjycus alvarezii*	MCX; MAX and HLB	[55,56,57,58]
92	11.24	1,2-Benzenedicarboxylic acid, butyl 2-ethylhexyl ester	Antioxidant, fungitoxic, cytotoxic activity, and antimicrobial agent drug development for arthritis and cancer	*P. chrysogenum*; twigs of *Thevetia peruviana* and root of *Plumbago zeylanica*; *Podophyllum hexandrum* rhizome *Alternaria alternata*	MCX; MAX	[55,59,60,61,62]
93		Phthalic acid, butyl hexyl ester	antimicrobial	*Boerhavia diffusa*, *Podophyllum hexandrum* rhizome	MCX; MAX	[62,63]
58	12,106	Benzo[h]quinoline, 2,4-dimethyl-; 2,4-Dimethylbenzo[h]quinoline	Antibacterial agent by inhibiting protein synthesis Antiprotozoal activity	*Lasiocarpa americana* Holm oak (*Quercus ilex* subsp. *ballota* (Desf.) Samp.) bark *Penicillium pusillum*	MCX; MAX and HLB	[64,65]
79	11.24	Benzeneethanamine, N-[(pentafluorophenyl)methylene]-.beta.,3,4-tris[(trimethylsilyl)oxy]-; N-(Pentafluorobenzylidene)-beta.,3,4-tris(trimethylsiloxy	Antimicrobial and antioxidant	*Boerhavia diffusa*	MCX; MAX and HLB	[63]
79	16.33	N-(Trifluoroacetyl)-O,O’,O’’-tris(trimethylsilyl)norepinephrine; N-(2-(3,4-Bis[(trimethylsilyl)oxy]phenyl)-2-[(trimethylsilyl)oxy]ethyl)-2,2,2-trifluoro	Schistosomicidal and molluscicidal activities	Extracts *Juniperus horizontalis* and uniperus communis L.	MCX; MAX	[66]
58	9.24	2,5-di-tert-Butyl-1,4-benzoquinone; 2,5-di-tert-Butyl-p-quinone; 2,5-Cyclohexadiene-1,4-dione, 2,5-bis(1,1-dimethylethyl)-; p-Benzoquinone	Not identified	*Phormidium tenue* and *Leptolyngbya*	MCX; MAX	[49,50,67]
61	13.499	Coumarin, 3,4-dihydro-4,5,7-trimethyl-; 4,5,7-Trimethyl-2-chromanone	anti-leishmaniasis agents; bacteriostatic and anti-tumor activity; anti-HIV; antifungal; antidiabetic	*Calophyllum lanigerum*; *Alternaria alternata*	MAX	[68,69,70]
66	24.10	2-Fluoro-5-trifluoromethylbenzoic acid, 3-hexadonic	antifungal	*P. chrosgenum*	HLB	[55,71]
70	32.64	L-Proline, N-valeryl-, heptadecyl ester	Antifungal against *A. alternata*	*Bacillus siamensis*, *Bacillus amyloliquefaciens*, and *Bacillus* *nakamurai Bacillus safensis* STJP	HLB	[72]
45	13.76	3,5-Dimethoxycinnamic acid; 2-Propenoic acid, 3-(3,5-dimethoxyphenyl)-; Cinnamic acid, 3,5-dimethoxy-(coumaric acid derivates)	Cytotoxic effect to human pepatoma tumor cells	*Wasebia japônica*, *Picrasma quassioides*, *Penicillium citreonigrum* XT20-134, *Sibiraea angustata*, *Verbesina gigantea*	HLB	[49,50,73,74]
77	18.18	1,2-Benzisothiazol-3-amine tbdms	Antioxidant and antimicrobial; antibacterial	Plants of *Thevetia neriifolia* *Acacia karoo* root extract	HLB	[56]
76	11.24	1,4-Bis(trimethylsilyl)benzene	Antimicrobial	*Acacia karoo* root	HLB	[56]
73	5.76	Oxime-, methoxy-phenyl	Antidotes for nerve agents, reactivate acetylcholinesterase	*Brassica juncea*	HLB	[75]
69	17.14	4-Ethylbenzoic acid, cyclopentyl ester	Not identified	*Pichia membranifaciens* Hmp-1 isolated from blackberry wine	HLB	[76]
76	16.25	2′,6′-Dihydroxyacetophenone,bis(trimethylsilyl) ether	Antimicrobial and antioxidant activity	*Camellia sinensis* and *Camellia assamica*	HLB	[77,78]
61	12.33	1,1,3,3,5,5,7,7-Octamethyl-7-(2-methylpropoxy) tetrasiloxan-1-ol	Not identified	*Ziziphus jujuba* mill.	HLB	[79]
58	10.68	Pentasiloxane, dodecamethyl; Dodecamethylpentasiloxane; 1,1,1,3,3,5,5,7,7,9,9,9-Dodecamethylpentasiloxane	Antimicrobial activity	*Solanum nigrum*	HLB	[80]
75	11.24	l-Leucine, N-cyclopropylcarbonyl-, heptadecyl ester	Antimicrobial activities	*Bacillus* species	HLB	[81]

## Data Availability

Not applicable.

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
