# Peer review of "Epigenetic Induction of Secondary Metabolites Production in Endophytic Fungi Penicillium chrysogenum and GC-MS Analysis of Crude Metabolites with Anti-HIV-1 Activity"

_microorganisms, 2023, doi:10.3390/microorganisms11061404_

Round 1

Reviewer 1 Report

In this manuscript, the authors isolated 5 endophytic fungi from Albizia adianthifolia and induced them to produce anti-HIV activities by adding some small epigenetic modifiers.At the same time, a fungus P. chrysogenum P03MB2 with high anti-HIV activity under sodium butyrate addition was found, and the secondary metabolites from active partially purified extracts were identified by GC-MS. This work provides experimental guidance for screening bio-control agents and discovering new bioactive compoundsHowever, I found that many descriptions of the results in the full text lacked experimental evidence support, and the descriptions of the results and figures were unclear. There are also usage error and awkward phrases throughout the manuscript that make it difficult to understand. I have provided more detailed items below for you to consider.

1.      Line 345: “these” should be “both them”.

2.      Lines 344-345: Where is the result of anti-HIV activities of five endophytic fungi?

3.      Line 347: Delete “P. chrysogenum.”; “The” should be “the”. Where is the result of BLAST?

4.      Please show the similarity of PO3PB2 and P03MB2 with Alternaria alternata and Penicillium chrysogenum, respectively.

5.      Lines 348-349: The results of this description are missing. Please confirm the full text carefully.

6.      Figure 1: The strain name should be italicized. Are the thresholds for the evolutionary branches of this phylogenetic tree all 100%? Please provide the distance of evolutionary branches.

7.      What does “TZM-bl” mean? Please provide the full name as it appears first in this article.

8.      Figure 2: Please place (%) after “cell viability”. Please show the abbreviation in the figure legend and please emphasize the biological duplications per treatment.

9.      Line 374: “shows” should be “showed”. Delete “among all treated and untreated extracts”. “P. chrysogenum strain P03MB2 extract showed……” should be “the extract of P. chrysogenum P03MB2 showed……”.

10.   This description in lines 374-376 is not be well understood. The “figure 3” in line 376 should not be bold.

11.   Please keep the strain name consistent throughout the text, eg., P. chrysogenum P03MB2 or P03MB2.

12.   Figure 3: Please place (%) after “HIV inhibition”. Please add error lines in this figure. What is the X-axis mean? Please explain in the figure legend.

13.   Figure 4C: A part of data information is lost. Please keep the style of figure consistent, such as abbreviations.

14.   Please provide the web of NIST library.

There are also usage error and awkward phrases throughout the manuscript that make it difficult to understand. I have provided more detailed items for you to consider.

Author Response

Reviewer 1

      In this manuscript, the authors isolated 5 endophytic fungi from Albizia adianthifolia and induced them to produce anti-HIV activities by adding some small epigenetic modifiers. At the same time, the fungus P. chrysogenum P03MB2 with high anti-HIV activity under sodium butyrate addition was found, and the secondary metabolites from active partially purified extracts were identified by GC-MS. This work provides experimental guidance for screening bio-control agents and discovering new bioactive compounds。 However, I found that many descriptions of the results in the full text lacked experimental evidence support, and the descriptions of the results and figures were unclear. There are also usage errors and awkward phrases throughout the manuscript that make it difficult to understand. I have provided more detailed items below for you to consider.

Response:

  1. Line 345: "These" should be "both them".

Response: The word “these” in line 345 has been removed and we rephrased the paragraph.

  1. Lines 344-345: Where is the result of anti-HIV activities of five endophytic fungi?

Response: The results of the anti-HIV activities are now incorporated as a supplementary Figure S1 and the tested isolates have been corrected as a total of four, not five as recorded in the submitted manuscript. Figure S1 shows the dose-response curve of anti-HIV-1 activities of four crude extracts treated with either valproic acid or sodium butyrate.

  1. Line 347: Delete “ chrysogenum.”; “The” should be “the”. Where is the result of BLAST?

Response: We thank the reviewer for noting this typographical error, we have revised the paragraph. The BLAST results have been incorporated as Panel A in Figure 1.

  1. Please show the similarity of PO3PL2 and P03MB2 with Alternaria alternata and Penicillium chrysogenum, respectively.

Response:  The similarity has been incorporated in Figure 1 as panel A. The phylogenetic tree showing the maximum likelihood of PO3MB2 is shown in Panel B.

  1. Lines 348-349: The results of this description are missing. Please confirm the full text carefully.

Response: The results described here have been incorporated as Supplementary Figure S1 (describing the initial screening of four fungal isolates for anti-HIV activity). The results show the anti-HIV results for the treated crude extracts (treated with either 25 μM valproic acid or 25 μM sodium butyrate) and AZT control.

  1. Figure 1: The strain name should be italicized. Are the thresholds for the evolutionary branches of this phylogenetic tree all 100%? Please provide the distance of evolutionary branches.

Response: The strain name has been italicized. The distance of evolutionary branches has been inserted.

  1. What does “TZM-bl” mean? Please provide the full name as it appears first in this article.

Responses: In lines 196-200, we have described the cell line in section 2.5 (Methods and Materials) and provided a rationale for choosing it in this study as follows “TZM-bl and HEK 293T are Hela cell lines that express large amounts of CD4 and CCR5. They were initially generated from JC. 53 cells by introducing separate integrated copies of the luciferase gene under the control of the HIV-1 promoter [37-39]. The TZM-bl cell line was chosen based on their high susceptibility to infection with most strains of HIV.

8.Figure 2: Please place (%) after “cell viability”. Please show the abbreviation in the figure legend and please emphasize the biological duplications per treatment.

Response: Figure 2 has been amended as suggested by the reviewer. The abbreviations are shown in Figure legend as follows: “AZT (Azidovudine), SB (sodium butyrate), PO3 (Plant 3- A. adianthifolia), M (Malt extract agar), P (Potato dextrose agar), B (Bark), 2 (isolate number), T (treated extract), NT (not treated extract).

9. Line 374: "Shows" should be "showed". Delete “among all treated and untreated extracts”. “ chrysogenum strain P03MB2 extract showed……” should be “the extract of P. chrysogenum P03MB2 showed……”.

Response: Thank you for the comment, the error has been corrected as suggested. The statement that reads “among all the treated and untreated extracts” has been deleted. It has been replaced with the following statement. In lines 384-386: “The P. chrysogenum P03MB2 crude extract showed anti-HIV activity with a half-maximal inhibitory concentration (IC₅₀) of 0.048 µg/mL with a fold change 5 compared to the untreated fungal extract (IC50 ) of 0.009 µg/mL (Figure 3)”.

10. This description in lines 374-376 is not well understood. The "Figure 3" in line 376 should not be bold.

Response: In line 383 the description has been changed as follows: “Antiviral effects of the crude extracts were measured by calculating the percentage of virus inhibition”.

In line 386: We have removed the bold emphasis on the Figure 3 reference in the paragraph and figure 3 has been placed at the end of the sentences in brackets.

11. Please keep the strain name consistent throughout the text, eg., chrysogenum P03MB2 or P03MB2.

Response: The strain name P03MB2 has been changed to P. chrysogenum PO3MB2 throughout the manuscript.

12. Figure 3: Please place (%) after “HIV inhibition”. Please add error lines in this figure. What is the X-axis mean? Please explain in the figure legend.

Response: In Figure 3: The percentage (%) has been moved to the front of the HIV inhibition (HIV inhibition as suggested and error lines has been placed. The meaning of the x-axis has been explained as "The x-axis showed serial dilution of crude extract concentrations in Log scale and y-axis showed the percentage of HIV-1 inhibition” and Figure legend explained as follows: Dose-response curve showing anti-HIV-1 activity of sodium butyrate-treated crude extracts of P. chrysogenum P03MB2T (blue) and untreated crude extracts of P. chrysogenum P03MB2NT (pink) tested in TZM-bl cell lines. Fungi-free sodium butyrate (SB) crude extract (green) was included as a negative control, while AZT (red) was used as a positive control. The graph shows inhibitory concentration at 50% inhibition (µg/mL).

Figure 4: Dose-response curves of three pre-concentration cartridges (A: HLB, B: MCX, and C: MAX all samples were eluted in 45% methanol) with sodium butyrate-treated (blue), untreated (pink), AZT (red) and sodium butyrate-inoculum free culture control (green). The x-axis showed serial dilution of crude extract concentrations in the log scale and the y-axis showed the percentage of HIV-1 inhibition. The anti-HIV activity of fractionated secondary fungal metabolites was tested via the treatment of TZM-bl cells. Inhibitory concentration at 50% inhibition (µg/mL) of treated P. chryosogenum (PO3MB2T) eluted in 45% Methanol in HLB is 0.6024 µg/ml compared to untreated crude extracts (PO3MB2NT) 5.053µg/ml and fold change of 8.39, MCX-45%: IC50 PO3MB2T= 4.246µg/ml; PO3MB2NT= 2.707µg/ml, MAX- 45% IC50: PO3MB2T= 1.456µg/ml; PO3MB2NT= 0.4809 µg/ml). The fold change of MCX and MAX eluted crude extracts were 0.63 and 0.33 respectively.

13.Figure 4C: A part of the data information is lost. Please keep the style of the figure consistent, such as abbreviations.

Response: We have changed the Figure to show the data for Panel C and ensured that there is consistency. We have corrected Figure 4 Legend as follows: Dose-response curves of three crude extract fractions (A: HLB, B: MCX, and C: MAX. All samples were eluted at 45% methanol with sodium butyrate-treated P. chrysogenum P03MB2T (blue), un-treated P. chrysogenum P03MB2NT (pink), AZT positive control (red), and fungi-free sodium butyrate-treated crude extract fraction (SB) as negative control (green). The x-axis shows serial dilution of crude extract concentrations in the Log scale and the y-axis shows the percentage of HIV-1 inhibition. The anti-HIV activity of fractionated secondary crude extracts was tested via the treatment of TZM-bl cells. In Panel A (HLB 45%: P. chrosogenum PO3MB2T, IC50 0.6024 µg/mL, P. chrysogenum PO3MB2NT, IC50 5.053 µg/mL); Panel B (MCX 45%: P. chrysogenum PO3MBT, IC50 4.24 6µg/mL; P. chrysogenum PO3MB2NT, IC50 2.707 µg/mL); Panel C (MAX 45%: P. chrysogenum PO3MB2T IC50 1.456 µg/mL, P. chrysogenum PO3MB2NT, IC50 0.4809 µg/mL).

14. Please provide the web of the NIST library.

Response: The webpage reference has been added in text and in the bibliography as follows: https://www.nist.gov/nist-research-library: Accessed on 12 October 2022.

Comments on the Quality of English Language

There are also usage errors and awkward phrases throughout the manuscript that make it difficult to understand. I have provided more detailed items for you to consider.

Response: Thank you for noting this, we have revised the manuscript to make it easy to read.

Reviewer 2 Report

First of all, I want to thank all the authors for this intersting paper

the idea is novel and sound applicable.

you assessed the effects of small molecular epigenetic modifiers in activating silent BGCs in endophytic fungi Penicillium chyrysogenum isolated from Albizia 103 adianthifolia followed by a comparative analysis of anti-HIV activity of produced secondary metabolites between treated and the untreated crude/fractionated extracts from the 105 fungal culture.

My question how u decside the comcentration of sodium butyrate, and valproic acid that can induce the epigentic change and how to adjust it to be sure of the repreducability if anyone will repate ur experiemnt to reach your results.

also you reach a an excellant stage by the GCMS and separte the most prominent compounds. But still dont reach the active fraction that acte as anti-HIV-1.

Also in vivo experiment will be essenial to close the circle. 

Good Luck and excellant work

medium

Author Response

Reviewer 2

Open Review

First of all, I want to thank all the authors for this interesting paper the idea is novel and sounds applicable.

You assessed the effects of small molecular epigenetic modifiers in activating silent BGCs in endophytic fungi Penicillium chrysogenum isolated from Albizia  adianthifolia followed by a comparative analysis of the anti-HIV activity of produced secondary metabolites between treated and the untreated crude/fractionated extracts from the 105 fungal culture.

Response: We thank the reviewer for the kind words and for noting the novelty provided by the manuscript.

My question is how u decide the concentration of sodium butyrate, and valproic acid that can induce the epigenetic change and how to adjust it to be sure of the reproducibility if anyone will repeat your experiment to reach your results.

Response: We first ran a training set where we examined a concentration range of epigenetic markers (valproic acid and sodium butyrate) on the inhibition of HIV-1 in TZM-bl cells. In literature, it is well recorded that a sub-inhibitory concentration (concentration below the minimum inhibitory concentration) of small molecular compounds (epigenetic inhibitors) differentially elicits the expression of biosynthetic gene clusters in microorganisms such as fungi. In this training set, we observed that 25 μM was the sub-inhibitory concentration that was not inhibitory to the cells nor caused any HIV-1 inhibition. In this way, we ensured that the observed inhibition was a direct effect of the compounds used (controls and crude extracts). We replicated this training set to ensure repeatability.

Also, you reach an excellent stage by the GCMS and separate the most prominent compounds. But still don’t reach the active fraction that acts as anti-HIV-1.

Response: We thank the reviewer for noting the importance of identifying the active compound from the fractions that results in a positive hit. The study reports on the initial phase of screening and only volatile compounds were considered for the reported study. In the future, we aim to comprehensively examine the available chemical space in the active fractions and collaborate with medicinal chemists to isolate and purify the active compounds.

Also in vivo, the experiment will be essential to close the circle. 

Response: We thank the reviewer for this important consideration. Since we have been able to show the potential of endophytic fungal isolates (isolated following ethnobotanical knowledge of medicinal plant bioactivities), the next step is to isolate and purify the active compounds, confirm in vitro bioactivities, and elucidate their mechanism of action. This information will be critical in designing experiments for animal studies (in vivo testing) such that we follow a specific target and reduce the number of animals for the experiments.

Good Luck and excellent work

Response: We thank the reviewer for the good wishes.

Reviewer 3 Report

The manuscript titled “Epigenetic induction of secondary metabolites production in endophytic fungi Penicillium chyrosogenum and GC-MS analysis of crude metabolites with anti-HIV-1 activity” by John P. Makhwitine et al.,

Title: Penicillium chyrosogenum should be in Italics.

Abstract: Partially purified extracts of treated P.chrysogenum showed 23 potent anti-HIV activity with an IC₅₀ of 0.6024 µg/mL compared to untreated 5.053 µg/mL with a 8- 24 fold increase.- Check the statement.

Line 82: Epigenetic modifiers are small molecular compounds that target the epigenetic processes such 83 as inhibiting the transfer of methyl, acetyl, or alkyl groups, thus leading to epigenetic alterations (Pillay et al. 2022; [21]).- Correct the citation and similarly throughout the manuscript.

Line 93: Exposure of Streptomyces coelicolor to sodium butyrate activated the biosynthetic pathway for the production of actinorhodin which was otherwise silent in the uninduced strain (Moore et al. 2012; 95 Zhu et al. 2014).- Check the Citation an use proper format.

One of the main points raised is that on what basis two elicidators were selected. I completely agree with the authors that certain BGC are not expressed in routine laboratory conditions. But Why authors did not try several different mediums to see the difference in metabolite production too. Even nutrient amendments can have drastic effect on metabolite production.

Justify the selection of 25 µM concentration of modulator. Different concentration should be used to record change in metabolite at different concentration range.

Provide the results for preliminary screening leading to the selection of one strain.

 Line 298- 301: Extraction part is not clear here. Authors are requested to explain it better. So, after incubating for 14 days, was the culture centrifuged and only mycelium extracted with methanol? If not why mycelium and cell free supernatant was not extracted separately?

Line 307-312: “The dried material was reconstituted 307 in 50% methanol (MeOH) to a concentration of 100 μg/mL, and 1 mL of this solution was spiked with 20 µL of phosphoric acid (Sigma-Aldrich, South Africa) and loaded onto HLB (hydrophilic-lipophilic-balanced reverse phase), MCX (Mixed-mode, strong Cation-eX- change), and MAX (Mixed mode, strong Anion-eXchange) cartridges obtained from Waters Corporation (Prague, Czech Republic).”

The active crude extract was fractionated using three different cartridges. Why not use just a simple basic C18 cartridge first.?

I don’t understand the logic of selecting only GC-MS. For untargeted metabolomics, it is also advised to include LC-MS/MS analysis to have a proper picture to understand the difference in metabolite production.

Figure 1: Provide all the binomial names in Italics.

No statistics was done in any of the assay results provided. Authors are advised to present the data with proper fitting statistics.

Genomic characterization of P. chrysogenum PB3MB2 should be done in order to identify and corelate the change in metabolite production by analysing the BGCs.

No specific comment on Language but very hard to understand. It is advised to rephrase bigger sentences into small ones for better understanding.

Author Response

The manuscript titled “Epigenetic induction of secondary metabolites production in endophytic fungi Penicillium chyrosogenum and GC-MS analysis of crude metabolites with anti-HIV-1 activity” by John P. Makhwitine et al.,

Title: Penicillium chyrosogenum should be in Italics.

Response: Thank you for noting this, The word “Penicillium chrysogenum” is now italicized in the title.

Abstract: Partially purified extracts of treated P.chrysogenum showed 23 potent anti-HIV activity with an IC₅₀ of 0.6024 µg/mL compared to untreated 5.053 µg/mL with an 8- 24 fold increase.- Check the statement.

Response: In line 23-26, the statement has been revised as follows: “Penicillium. chrysogenum PO3MB2 showed anti-HIV activity with an IC₅₀ of 0.6024 µg/mL compared to untreated fungal crude extract (IC50 5.053 µg/mL) when treated with sodium butyrate”.

Line 82: Epigenetic modifiers are small molecular compounds that target the epigenetic processes such 83 as inhibiting the transfer of methyl, acetyl, or alkyl groups, thus leading to epigenetic alterations (Pillay et al. 2022; [21]).- Correct the citation and similarly throughout the manuscript.

Response: In line 85: The citation has been corrected and all references checked for uniformity throughout the manuscript.

Line 93: Exposure of Streptomyces coelicolor to sodium butyrate activated the biosynthetic pathway for the production of actinorhodin which was otherwise silent in the uninduced strain (Moore et al. 2012; 95 Zhu et al. 2014).- Check the Citation and use the proper format.

Response: In line 94: The citation has been formatted accordingly.

One of the main points raised is on what basis two elucidators were selected. I completely agree with the authors that certain BGCs are not expressed in routine laboratory conditions. But Why authors did not try several different mediums to see the difference in metabolite production too? Even nutrient amendments can have a drastic effect on metabolite production.

Response: We note that the reviewer recommends that we also consider the systematic variation of growth conditions and media components (OSMAC approach). Indeed, this has been proven as one of the many cultivation-based strategies (epigenetic induction, co-culture, OSMAC, etc) to induce silent biosynthetic gene clusters. The OSMAC approach (epigenetic modifiers in combination with various solid and liquid media) is an important consideration that in our lab we are actively pursuing to improve the yield for the isolation and purification of bioactive compounds from this isolate.

Justify the selection of 25 µM concentration of modulator. Different concentrations should be used to record changes in metabolite at different concentration ranges.

Response: We first ran a training set where we examined a concentration range of epigenetic markers (valproic acid and sodium butyrate) on the inhibition of HIV-1 in TZM-bl cells. In literature, it is well recorded that a sub-inhibitory concentration (concentration below the minimum inhibitory concentration) of small molecular compounds (epigenetic inhibitors) differentially elicits the expression of biosynthetic gene clusters in microorganisms such as fungi. In this training set, we observed that 25 μM was the sub-inhibitory concentration that was not inhibitory to the cells nor caused any HIV-1 inhibition. In this way, we ensured that the observed inhibition was a direct effect of the compounds used (controls and crude extracts). We replicated this training set to ensure repeatability.

In summary, the concentration was selected on the basis that it does not contribute to the observed HIV-1 inhibition either by inhibiting the TZM-bl cells or having a direct effect on the HIV-1 inhibition.

Provide the results for preliminary screening leading to the selection of one strain. (Please the supplementary figure)

Response: The results of the initial antiviral screening of the four different endophytic fungal crude extracts have been included as supplementary Figure S1.

Line 298- 301: The extraction part is not clear here. Authors are requested to explain it better. So, after incubating for 14 days, was the culture centrifuged and only mycelium extracted with methanol? If not why mycelium and cell-free supernatant was not extracted separately?

Response: After 14 days of incubation, an equal volume of methanol was added to the whole culture (broth and mycelia) and incubated with shaking overnight. Thereafter, the mycelia were pressed to remove all the liquid parts back to the supernatant, and the mycelia were discarded. We did some trials to check if we get activity from mycelia after this extraction (resuspending the pressed mycelia in methanol and re-extraction) and found that there was no residual activity. We have rephrased the manuscript in lines 142-150 as follows: “After incubation, an equal volume of absolute methanol (ChemLab supplies, South Africa) was added to the whole fungal culture and incubated overnight with shaking on a rotary shaker (Reflecta Laboratory supplies, South Africa) at 150 rpm. The mycelia were then separated using gauze and the culture supernatant was transferred into a pre-weighed. The retained supernatant was evaporated at 40°C to dry the extracts. The dried crude extracts were stored at -80⁰C until further use. Before each use, the extracts were resuspended in distilled water to a concentration of 300 μg/mL”.

Line 307-312: “The dried material was reconstituted 307 in 50% methanol (MeOH) to a concentration of 100 μg/mL, and 1 mL of this solution was spiked with 20 µL of phosphoric acid (Sigma-Aldrich, South Africa) and loaded onto HLB (hydrophilic-lipophilic-balanced reverse phase), MCX (Mixed-mode, strong Cation-eX- change), and MAX (Mixed mode, strong Anion-eXchange) cartridges obtained from Waters Corporation (Prague, Czech Republic).”

The active crude extract was fractionated using three different cartridges. Why not use just a simple basic C18 cartridge first.?

Response: In line 520-527 we have justified as follows, “The adoption of the three cartridges was inspired by a study published by Stoszcko et al. [45] where they reported on the discovery of gliotoxin with HIV-1 latency reversal properties. The Oasis cartridges are the next generation of C18 reverse-phase prepacked columns with improved chemistry. They are designed for the extraction of a wide range of compound chemistries from various matrices. In addition, they provide excellent recoveries without the need of worrying about the drying of the sorbent matrix. We chose these cartridges based on their suitability for an untargeted discovery approach where we do not know the chemical properties of the bioactive compound”.

I don’t understand the logic of selecting only GC-MS. For untargeted metabolomics, it is also advised to include LC-MS/MS analysis to have a proper picture to understand the difference in metabolite production.

Response: We thank the reviewer for this important consideration, and we agree that LCMS analysis will provide a comprehensive chemical space detail. The limitation in our study is that we only analyzed volatile compounds using GCMS and we do consider that some bioactive compounds might have been missed. The basis of our choice is that some studies have shown that GCMS can also be a powerful tool for elucidating bioactive metabolites. We are considering using LCMS analysis in future studies.

In line 542-546 we have justified the use of GCMS as follows, “The limitation of using GC-MS is that it only identifies the volatile compound and its possible that other compounds group may be responsible for the observed bioactivities. In future, high resolution liquid chromatography-tandem mass spectrometry (HRLCMS/MS) should be considered to comprehensively cover the chemical space of compounds that might be responsible for the observed bioactivities.

Figure 1: Provide all the binomial names in Italics.

Response: The italics have been included in Figure 1, accordingly.

No statistics were done in any of the assay results provided. Authors are advised to present the data with proper fitting statistics.

 Response: The results of all the experiments were averages of three independent experiments GraphPad Prism was used to analyse the concentration -response data of anti-HIV-1 activities with non-linear curve fitting programme to determine IC50 and CC50 values. The percentage of HIV-1 inhibition of the crude extracts was determined using the following formula:  

Genomic characterization of P. chrysogenum PB3MB2 should be done to identify and correlate the change in metabolite production by analyzing the BGCs.

Response: We thank the reviewer for this important comment. In this study, we report on the initial screening, and we emphasize using the epigenetic modifiers in activating BGCs that otherwise are silent when fungal isolates are cultivated in a laboratory environment. This study will indeed form the basis for functional genomics studies where we try to link the bioactive metabolites to their biosynthetic genes and also look at the differential expression of these biosynthetic pathways upon various environmental or chemical perturbations.

In line 550-553 this comment is justified as: However, more rounds of purification of the bioactive extracts and deep genome analysis of fungal species are recommended to identify the compound responsible for bioactivity and to gain insight into the differential expression of the encoding biosynthetic pathway upon perturbation by sodium butyrate.

Round 2

Reviewer 3 Report

The manuscript has improved substantially but still has some minor issues, which need to be answered before endorsing it for publication:

The authors wrote that "The dried crude extracts were stored at -80⁰C until further use. Before each use, the extracts were resuspended in distilled water to a concentration of 300 μg/mL”. How can the authors be sure that methanolic extracts, when resuspended in water can dissolve all the molecules present in the crude extract?

Author Response

The authors wrote that "The dried crude extracts were stored at -80⁰C until further use. Before each use, the extracts were resuspended in distilled water to a concentration of 300 μg/mL”. How can the authors be sure that methanolic extracts, when resuspended in water can dissolve all the molecules present in the crude extract?

Response: Thank you for this question, we first tried different solvent formulations including distilled water, 0.2% DMSO and 50% methanol. The 50% methanol interfered with bioactivity while 0.2% DMSO dissolved extracts showed less activity than the extracts dissolved in water. We then decided to use water as our preferred solvent for bioactivity screening. We also considered that our extracts completely dissolved in water suggesting that the compounds were water soluble. After fractionation, the 45% HLB fraction showed more activity. The suggested that our bioactive compound(s) might be hydrophilic (tend to easily dissolve in water) since the HLB matrix is based on a hydrophilic-lipophilic balanced reverse-phase matrix.